# Artificial Intelligence-Augmented Propensity Score, Cost Effectiveness and Computational Ethical Analysis of Cardiac Arrest and Active Cancer with Novel Mortality Predictive Score

**DOI:** 10.3390/medicina58081039

**Published:** 2022-08-03

**Authors:** Dominique J. Monlezun, Oleg Sinyavskiy, Nathaniel Peters, Lorraine Steigner, Timothy Aksamit, Maria Ines Girault, Alberto Garcia, Colleen Gallagher, Cezar Iliescu

**Affiliations:** 1Department of Cardiology, The University of Texas MD Anderson Cancer Center, Houston, TX 77030, USA; ciliescu@mdanderson.org; 2UNESCO Chair in Bioethics & Human Rights, 00163 Rome, Italy; agg@arcol.org (A.G.); cmgallagher@mdanderson.org (C.G.); 3School of Bioethics, Universidad Anahuac México, Mexico City 52786, Mexico; maria.girault@anahuac.mx; 4Center for Artificial Intelligence and Health Equities, Global System Analytics & Structures, New Orleans, LA 70112, USA; nathanieljpeters5@gmail.com (N.P.); llsteigner@gmail.com (L.S.); 5Department of Public Health, Asfendiyarov Kazakh National Medical University, Almaty 050000, Kazakhstan; osinyavsky@yahoo.com; 6Department of Pulmonary Medicine, Mayo Clinic, Rochester, MN 55905, USA; aksamit.timothy@mayo.edu; 7Pontifical Academy for Life, 00193 Rome, Italy; 8Section of Integrated Ethics, The University of Texas MD Anderson Cancer Center, Houston, TX 77030, USA

**Keywords:** artificial intelligence, cardiac arrest, cardio-oncology, cost effectiveness, ethics, equity

## Abstract

*Background and objectives*: Little is known about outcome improvements and disparities in cardiac arrest and active cancer. We performed the first known AI and propensity score (PS)-augmented clinical, cost-effectiveness, and computational ethical analysis of cardio-oncology cardiac arrests including left heart catheterization (LHC)-related mortality reduction and related disparities. *Materials and methods*: A nationally representative cohort analysis was performed for mortality and cost by active cancer using the largest United States all-payer inpatient dataset, the National Inpatient Sample, from 2016 to 2018, using deep learning and machine learning augmented propensity score-adjusted (ML-PS) multivariable regression which informed cost-effectiveness and ethical analyses. The Cardiac Arrest Cardio-Oncology Score (CACOS) was then created for the above population and validated. The results informed the computational ethical analysis to determine ethical and related policy recommendations. *Results*: Of the 101,521,656 hospitalizations, 6,656,883 (6.56%) suffered cardiac arrest of whom 61,300 (0.92%) had active cancer. Patients with versus without active cancer were significantly less likely to receive an inpatient LHC (7.42% versus 20.79%, *p* < 0.001). In ML-PS regression in active cancer, post-arrest LHC significantly reduced mortality (OR 0.18, 95%CI 0.14–0.24, *p* < 0.001) which PS matching confirmed by up to 42.87% (95%CI 35.56–50.18, *p* < 0.001). The CACOS model included the predictors of no inpatient LHC, PEA initial rhythm, metastatic malignancy, and high-risk malignancy (leukemia, pancreas, liver, biliary, and lung). Cost-benefit analysis indicated 292 racial minorities and $2.16 billion could be saved annually by reducing racial disparities in LHC. Ethical analysis indicated the convergent consensus across diverse belief systems that such disparities should be eliminated to optimize just and equitable outcomes. *Conclusions*: This AI-guided empirical and ethical analysis provides a novel demonstration of LHC mortality reductions in cardio-oncology cardiac arrest and related disparities, along with an innovative predictive model that can be integrated within the digital ecosystem of modern healthcare systems to improve equitable clinical and public health outcomes.

## 1. Introduction

Cardiac arrest (or the sudden loss of cardiac function resulting in complete cessation of blood flow throughout the body) accounts for up to 11.1 million or 20% of global deaths, including half of the deaths from cardiovascular disease (the top medical cause of mortality) [1,2,3]. It carries a 92.3% 1-year mortality rate (nearly unchanged for the last 22 years) and includes limited to no robust data for most middle to low-income nations particularly in Africa, South America, central and south Asia, and the Middle East [1,2,3]. Every minute the post-arrest heart fails to pump blood to the brain, 1.9 million neurons die [4,5,6]. This translates into the optimal cut-off time for cardiopulmonary resuscitation (CPR) being approximately 20 min for 90% of patients, after which the likelihood of favorable neurological outcome significantly falls. Recent research suggests this cut-off time is as early as 12 min, meaning the vast majority of patients die before arriving at the hospital, and even then survival remains low despite aggressive interventions [4,5,6]. The high inpatient and rehabilitation direct costs per person for cardiac arrest is $102,017 with a total cost of cardiac arrests in the United States (US) alone exceeding $33 billion annually [7,8]. Post-arrest inpatient care is often resource-intensive, both in acute care initially upon admission followed by an often-extended course of rehabilitation and outpatient follow-up care (with both settings and intensity of resources being more difficult to access in lower-income communities, healthcare systems, and nations) [7,8].

The vast majority of cardiac arrests occur out of hospital, disproportionately impact socially disadvantaged groups (including racial minority, lower income, and underinsured to no insurance sub-groups), and often result from the culmination of chronically sub-optimally controlled medical conditions (including in socially disadvantaged groups having higher barriers to healthcare system access and related prevention, diagnosis, and treatment of conditions), and remain plagued by persistent health disparities (of incidence, treatment, and survival) [9,10]. When too many conditions build up over too many years, the end result is that the patient’s heart can simply give up. When there are too many potential failures in the equitable distribution of optimal healthcare system performance for patients (regardless of their socially disadvantaged or advantaged groups alignment), too many people’s hearts may fail. As such, cardiac arrest may represent one of the most clinically sensitive and unified markers of healthcare system performance and disparity. Health disparities (or inequalities) refer to the differences in health outcomes for socially disadvantaged groups (specified above) following systematic disadvantages and even discrimination [11]. They contribute to worse health and health risks than other social groups (whose greater social advantage or relative position in latent social hierarchies typically confers greater resiliency (to social and personal shocks_ and social mobility and influence to pursue one’s ends (through greater prestige, power, and/or wealth)). Health equity is increased when such disparities are decreased. Cardiac arrest, therefore, may be a high yield target for improved health policy and system performance yielding outsized clinical, financial, and societal net benefits for those socially disadvantaged groups who stand to gain more from such benefits. The need to sufficiently address this target is accentuated particularly among patients with cancer. Given how it represents the second leading medical cause of death, the co-prevalence of cardiovascular disease and cancer (or cardio-oncology) is only expected to grow (with a globally aging population and thus the related co-prevalence of cardiac arrest). Minimal studies within cardiac arrest research have particularly addressed cancer, and significant baseline and outcome differences in arrest appear to occur in patients with concurrent cancer [12,13].

Yet equitably and effectively improving cardio-oncology cardiac arrest incidence and outcomes are challenged by notable limitations in the existing research literature (including a limited number of studies in the last two decades or robust methodologies). Such studies on cardiac arrest and cancer have relied on small samples from a small number of sites, failed to investigate interventions that may improve outcomes (and by extension none utilized causal inference analysis of such interventions), had little to no granular assessment by primary malignancy (of anatomic origin), and none featured any substantive disparity analysis, developed any clinical prediction tool to risk stratify patients, deployed cost-effectiveness analysis to identify optimal interventions, or utilized any form of artificial intelligence (AI) including machine learning (ML) or deep learning (DL) [14,15,16,17,18,19]. The widespread absence of risk prediction models and of AI in cardio-oncology cardiac arrest is a particularly stark challenge given the lack of sustained and substantive progress in equitable and effective cardiac arrest prevention and treatment (that historically have relied on providers’ clinical judgement and traditional statistics). Implantable cardioverter devices (ICDs) can reliably prevent ventricular arrhythmia-related cardiac arrest for patients meeting the current clinical criteria for ICDs (left ventricular ejection fraction < 30%), though this only covers 20% of all cardiac arrest cases [20]. Thus, at least 81 clinical predictive models (though not specific for cancer) have been developed to provide more precise risk stratification for prevention [21], yet the sole reliance of such models on traditional statistics limits their accuracy, precision, real-time adaptability, and clinical utility (as they typically identify subgroup risk at finite time points that typically exclude rarer prediction outliers while also failing to adapt with new data) [22,23,24]. A 2022 *Nature* study, therefore, utilized an integration of DL neural networks to accurately predict individual-specific survival curves up to 10 years out using only raw images from contrast-enhanced cardiac magnetic resonance (outperforming traditional survival models and clinical variables) [25]. Such technical effectiveness of AI-based approaches holds a significant (and plausible) potential to efficiently and equitably improve cardiac arrest outcomes (such a model can be integrated with diverse healthcare systems’ EHRs to provide real-time clinical decision support tools, particularly for systems with less material and specialty resources to better identify higher risk patients who need more aggressive prevention).

This study, therefore, sought to provide the first comprehensive (with integrated clinical, cost, and ethical) analysis of cardio-oncology cardiac arrests to optimize clinical effectiveness, cost efficiency, and health equity (for an optimal sustainable net benefit of patients and populations within the context of modern healthcare systems). It is thus the first known study encompassing the following: (a) production of a cardiac arrest mortality prediction score among patients with active cancer; (b) two separate validations in two separate populations of that score; (c) propensity score analysis of interventions reducing mortality in cardio-oncology cardiac arrest; (d) multi-center nationally representative epidemiologic analysis of patients with cardio-oncology and cardiac arrest identifying mortality risk and protective factors; (e) accomplishing the above based on primary malignancy both solid and non-solid in addition to active versus prior and metastatic versus non-metastatic malignancy; (e) the most recent available nationally representative data including up to 2018; (f) race, income, and insurance disparity analysis of cardio-oncological cardiac arrest prevalence, procedures, and outcomes; (g) longitudinal analysis of cardiac arrest with active cancer by procedural interventions; (h) regional analysis of the above; and (i) computational ethical analysis of the above. Additionally, this study is the largest known to date for patients with active cancer and cardiac arrest.

## 2. Materials and Methods

### 2.1. Data Source

The data source for this study is the largest publicly available U.S. all-payer inpatient healthcare administrative dataset, the National Inpatient Sample (NIS), sponsored by the Agency for Healthcare Research and Quality (AHRQ) within the US Department of Health and Human Services (DHHS) [26]. From 2016 onward, the NIS adopted the International Classification of Diseases, Tenth Revision, Clinical Modification (ICD-10-CM). The dataset includes demographic, comorbidity, procedural, complication, mortality, length of stay, total cost, and hospital characteristics for each hospitalization. The 2016, 2017, and 2018 NIS datasets were selected for this study as they are among the latest available datasets and the first to use ICD-10 coding and so better reflect current clinical trends in diagnoses, treatments, and outcomes compared to prior years. The selection of an inpatient dataset more generally was chosen given the global shortage of reliable data on out-of-hospital cardiac arrest and longitudinal follow-up. Thus, this study sought to help provide an initial comprehensive description of the current knowledge of cardio-oncology cardiac arrest using the more reliable inpatient data collection through EHR-based ICD10 coding. Study inclusion criteria included all NIS hospitalizations for adults aged 18 years or older during the above index time periods. Per the US DHHS and National Bureau of Economic Research, no review by an Institutional Review Board (IRB) is required for the NIS under the HIPPA Privacy Rule since the NIS is a limited data set (in which 16 direct identified specified by the privacy Rule have been removed) [27,28]. This study used de-identified data and was conducted according to the ethical principles in the Declaration of Helsinki.

### 2.2. Study Design

To conduct a more comprehensive analysis more broadly and practically applicable within current healthcare systems, the primary analysis consisted of AI-driven Computational Ethics and policy analysis (AiCE) according to its first empirical (clinical then economic) step then the second ethical-policy step [29,30]. This ethically aligned co-design of trustworthy AI methodology [31] features the patient-focused transparent integration of scientific and ethical methodologies to generate reliable, robust, and equitable results. The ultimate strategic aim is to therefore optimize value-based healthcare at the global population and individual levels, created through the end-to-end collaboration of clinicians, data scientists, healthcare system leaders, policy-makers, and community members [29].

The first empirical step featured a nationally representative retrospective longitudinal multicenter cohort analysis of inpatient mortality and total cost among all hospitalized adults. It additionally included a nested sub-group analysis among patients with cardiac arrest, according to the presence or absence of active cancer and the presence or absence of left heart catheterization (LHC) using Machine Learning-augmented Propensity Score adjusted multivariable regression (ML-PSr) and DL artificial neural network. These analyses informed the creation and validation of the novel Cardiac Arrest Cardio-Oncology Score (CACOS), the first known clinical predictive model of inpatient mortality following a sustained return of spontaneous circulation (ROSC) after cardiac arrest in active cancer (and by extension the first to be AI-augmented). A cost-effectiveness and benefit analysis was then conducted using the above clinical results. This empirical step was followed by the final ethical-policy step in which the above AI-augmented empirical results informed a pluralistic-based global bioethical analysis to optimize equitable care for the above patient populations. This integrated analytic approach of AiCE was additionally chosen given the less substantive and sustainable impact that prior studies (focusing on clinical, economic, or ethical dimensions separately) have on improving healthcare policy and healthcare systems. Despite the 2001 US Institute of Medicine’s *Crossing the Quality Chasm* and 2002 *Unequal Treatment* advocating that healthcare systems deliver quality healthcare (that is safe, efficient, effective, timely, patient-centered, and equitable) with 20 years of progress on the first five aims, healthcare equity notably trails the others [32]. This helps underline the importance of such methodological innovations as AiCE as an innovative AI-accelerated end-to-end translational research methodology with a value-healthcare orientation.

### 2.3. Descriptive and Bivariable Statistical Analysis

Descriptive statistics were performed for the full sample to define the prevalence of cardiac arrest among all adult hospitalizations. Sub-group analysis was then performed among patients with cardiac arrest featuring bivariable analysis by active cancer (yes/no) across the full 2016–2018 duration and within each year separately (2016, 2017, and 2018). For continuous variables, independent sample t-tests were performed to compare means, and Wilcoxon rank sum tests were performed for medians. For categorical variables, Pearson Chi-square tests or Fisher exact tests were performed to compare proportions as applicable.

Demographics included age, sex, race, income, insurance, urban density, and region. Comorbidities were selected for analysis (and identified in the dataset by their ICD-10 codes) based on their clinical and/or statistical significance identified in prior published studies and current clinical practice. They included cancer status (active, prior, metastatic, and prior chemotherapy, radiation, and immunotherapy), hypertension, coronary artery disease, atrial fibrillation, congestive heart failure, cirrhosis, and chronic kidney disease. Inpatient interventions included left heart catheterization (LHC) and percutaneous coronary intervention (PCI). The 26 primary malignancies investigated included brain and nervous system, head or neck, thyroid, breast, lung, esophagus, stomach, pancreas, liver or bile system, rectum or anus, colon, peritoneum, bone or connective tissue system, hematological malignancies (including Hodgkin lymphoma, non-Hodgkin lymphoma, leukemia, and multiple myeloma), melanoma, non-melanoma skin, uterus, cervix, ovarian, prostate, testes, bladder, and renal.

### 2.4. Regression Statistical Analysis, Machine Learning Analysis, and Model Optimization Overview

The primary outcome was inpatient mortality (yes/no), and the secondary outcomes were the length of stay (LOS) in days and total cost (in U.S. dollars [$]).

To maximize the likelihood of internally and externally valid and replicable results, a regression model performance was optimized according to the following sequential process. First, variables that were clinically or statistically significant were identified in the existing literature, clinical practice, and bivariable analysis to be considered in the final regression models. Second, those variables were included in the forward and backward stepwise regression to augment decision-making on the variables ultimately included in the final regression models. Third, the regression results were compared to those generated by backward propagation neural network ML to ensure comparability by root mean squared error and accuracy. Fourth, the regression model performance was additionally assessed with a correlation matrix, the area under the curve, Hosmer–Lemeshow goodness-of-fit test, Akaike and Schwarz Bayesian information criterion, variance inflation factor, and tolerance, multicollinearity, and specification error. Fifth, the models were iteratively run to fine-tune models until the above process confirmed optimal performance with the final models and included variables.

### 2.5. Machine Learning-Augmented Propensity Score Adjusted Multivariable Regression

Regression analysis featured the particular technique of ML-PSr [33,34,35], performed on the above NIS dataset for the reasons listed above. The propensity score for the likelihood of undergoing inpatient LHC (the treatment) was first created (utilizing the same above variables used in the final regression model given the double propensity score adjustment method), a balance was confirmed among blocks, and then the propensity score was included in the final regression models as an adjusted variable [36,37]. This causal inference approach of propensity score adjustment was selected because it is a widely accepted methodology to reduce (but cannot eliminate) selection bias and the effect of confounding variables in non-randomized observational studies. Competing causal inference approaches including fixed, random, and mixed effects were not appropriate (though these methods have the added advantage of reducing unobserved variable bias) because the NIS, by design, lacked adequate repeated hospitalizations from the same subjects. A propensity score adjustment was used, rather than a simple covariate adjustment (without the propensity score), to enable a more complex propensity score model (i.e., able to test interactions and higher order terms to produce the best-estimated probability of treatment assignment) without risking over-parameterizing, while still permitting diagnostic analysis of the final models to be done to confirm superior performance over a simple covariate adjustment without the propensity score. Finally, the propensity score adjustment rather than competing propensity score techniques was used because of its superior performance in the appropriate context (confirmed by current statistical theory and adequate diagnostic quantitative testing of the final models in cardiovascular studies) [36,37], and because its inclusion in the final regression models had sufficient performance confirmation by the above the specified optimization process.

The above analysis was augmented by ML to confirm adequately robust estimates by amplifying the assumptions 1000-fold for each algorithm, re-running the model using the below algorithms, and collapsing the results into stable mean results to confirm the above traditional statistical analysis. A total of 43 supervised learning algorithms were utilized with 10-fold cross-validations selected based upon the data type to determine the top performing. Performances among algorithms were assessed based on higher accuracy, lower root relative squared error (RRSE) with model acceptability set at 100% (for comparison among ML algorithms), and lower root mean squared error (RMSE, for comparison to traditional statistical results). The following algorithms were tested: Bayes Net, naïve Bayes, naïve Bayes multinomial text, and naïve Bayes updateable, logistic, multilayer perceptron, stochastic gradient descent, stochastic gradient descent text, simple logistic, sequential minimal optimization, voted perceptron, instance-bases learning with parameter K-start, locally weighted learning, adaptive boosting (AdaBoostM1), attribute selected classifier, bagging, classification via regression, cross-validation parameter selection, iterative classifier optimizer, logit boost, multiclass classifier, multiclass classifier updateable, multi-scheme, random committee, randomizable filtered classifier, random sub-space, stacking, vote, weighted instances handler wrapper, input mapped classifier, decision table, repeated incremental pruning to produce error reduction (RIPPER), one rule, part rule, zero rule, decision stump, Hoeffding tree, J48, logistic model tree, random forest, random tree, and reduced error pruning tree.

The utility of this above hybrid analytic approach (including the rationale for algorithm selection), which integrates the traditional statistical method of frequentist-based multivariable regression (supported by propensity score-based causal inference analysis) and supervised learning-based ML, has been previously demonstrated [38,39,40,41,42,43]. Causal inference results which are more familiar to medical science audiences can be confirmed and replicated automatically through ML, which has the added advantage of being integrated with EHRs (and thus may accelerate real-time results to guide clinical and organization decisions on larger high-dimensional datasets as they already increasingly do for other economic sectors outside of medicine) while producing more rapid and accurate results compared to traditional statistics in carefully defined contexts.

### 2.6. Propensity Score Matching

Propensity score matching (PSM) was conducted to estimate the average treatment effect (ATE) for LHC using the same variables identified in the final above model for ML-PSr for mortality. This was done to compare results to the post-regression marginal effect (given the familiarity, popularity, and ease of interpretation of this technique for clinical audiences and to allow more robust analysis across diverse techniques of the possible association between mortality and LHC among post-arrest patients with active cancer).

### 2.7. Health Equity Analysis

Health equity analysis was conducted to assess the divergence between observed values (for cardiac arrest prevalence, treatment, and outcomes) from predicted values (based on population distribution described by the latest 2021 US census data).

### 2.8. Deep Learning

DL analysis was conducted with neural network backpropagation in which the network nodes were set at sigmoid, and iterative model optimization was conducted to determine the appropriate number of hidden layers and nodes [44]. DL was deployed for the primary outcome of mortality with the same variables selected for the above statistical model of ML-PSr of mortality by LHC. The model configurations included a learning rate of 0.3, a momentum rate of 0.2, 500 epochs, an 80% split of the original dataset for the training sub-set, 20% for the validation sub-set, and a threshold of 20 for consecutive errors. Given the NIS dataset advantages listed above, no comparable available datasets were deemed sufficient to allow a model comparison, so untested datasets are forthcoming (including the forthcoming 2019 and 2020 datasets).

### 2.9. CACOS Predictive Model Creation, Validation, and Calibration

An AI-augmented clinical predictive model of inpatient mortality among post-arrest patients with active cancer was then constructed and validated. To optimize early clinical utility, sensitivity was prioritized (over specificity) using only ICD10 codes that could be extracted from EHRs on hospital admission to improve clinical support decision tools through more rapid risk stratification. This was meant to allow more precise identification of patients likely to ultimately survive and thus benefit from initial aggressive treatments (particularly given the prevalent social disparities in arrest treatment as noted by Rivera, 2018). Accordingly, for the model, the 2016 NIS sample was divided in a 1:1 ratio into a derivation sub-sample and a verification sub-sample. Bivariable analysis, forward and backward stepwise logistic regression, and ML-PS regression with ML-augmented model performance optimization (along with the above model diagnostic tests) were performed. These techniques were utilized to identify variables independently and significantly associated with inpatient mortality after sustained ROSC in active cancer in the derivation sub-sample to generate variable candidates for the final CACOS model. Once peak performance was confirmed, the nearest whole number was assigned to the independent variables from the final regression model to create the ultimate version of CACOS.

This predictive model was then assessed in both the derivation and verification sub-samples before being validated twice in two separate samples (both the 2017 and 2018 NIS datasets). According to the AHQR, the vast majority of hospitalizations are from unique individuals (rather than readmissions), though it is not possible due to the de-identified nature of the data to track longitudinally at the individual patient level [45]. Therefore, the three separate NIS samples from 2016, 2017, and 2018 were considered independent and external to each other overall for the purposes of model generation and validation.

Calibration plots were created for CACOS in the 2016 NIS derivation and verification sub-samples with the 45-degree dotted line representing perfect calibration (with equal predicted and observed probabilities), each dot representing sequential tenths of the sample, and the blue line representing the smooth Locally Weighted Scatterplot Smoothing (LOWESS) line [46]. Given the Hosmer–Lemeshow test’s strong sample size dependency and function simply as an overall calibration measure (without clear superiority to the calibration plot), it was not used as an additional measure of calibration [47].

### 2.10. Cost-Effectiveness and Cost Benefit Analysis

Cost-effectiveness analysis (CEA) was conducted according to the commonly accepted methodology described by the US Centers for Disease Control and Prevention (CDC): the net cost of the intervention (implementation cost minus the averted cost) divided by the change in health outcomes [48]. The model inputs include the following. LHC was the intervention and implementation costs independently attributed to inpatient LHC according to ML-PS multivariable regression among patients with cardiac arrest and active cancer ($11,643.48). The averted cost was set by the most recently available statistical value of human life as described by the US federal government ($7.4 million) [49]. The change in health outcome was the PSM ATE mortality reduction with LHC.

Cost benefit analysis (CBA) was also performed according to the commonly accepted methodology as described by the CDC: cost minus benefit, or the above implementation costs (LHC cost multiplied by the number of patients with cardiac arrest and active cancer who received LHC) minus the statistical value of human lives saved, multiplied by the number of patients who received LHC [50].

### 2.11. Ethical and Policy Analysis

The second or ethical-policy step within AiCE was then conducted by integrating the above quantitative analyses with ethical analysis using the pluralistic global bioethical framework of the Personalist Social Contract (PSC) [51,52,53,54]. The PSC is a novel integration of modern ethics (principally utilitarianism-informed Rawlsian social contract of political liberalism, bounded by Kantian deontology and informed by feminist, Marxist, deconstructionist, and ecological ethics) and classical ethics (principally Thomistic-Aristotelian virtue ethics, articulated by William Carlo’s *esse*/essence revision of Norris Clarke’s Strong Thomistic Personalism, a derivative formulation of Thomism as a development of classical Aristotelianism) [55,56,57,58,59,60,61].

The PSC was chosen as the primary ethical framework for its (a) practical, (b) political, and (c) philosophical advantages over competing frameworks. (a) Practically, it is historically articulated in (and makes philosophically intelligible) the world’s most dominant and cited ethical system (of human dignity-based rights and duties) as expressed paradigmatically by the UN’s 1948 United Nations Declaration of Human Rights (UDHR) and derivative system of modern international law and related international ethical conventions. (b) Politically, it substantively accounts for and can facilitate the convergence of the world’s nations (including through the UN explicitly grounded in the UDHR) and belief systems (including the above) on shared ethical conclusions as historically demonstrated since the modern world (united at the end of World War II to prevent future such catastrophic world wars). (c) Philosophically, it avoids the foundational metaphysical weaknesses (and resultant logical self-contradictions and struggles for deriving ethical conclusions within modern ethics) through the classic Aristotelian-derived Thomism and its Thomistic Personalist formulation. Yet it is made more intelligible in modern terms (emphasizing the centrality of each individual as a person subjectively experiencing an objective reality). It additionally produces the conclusions that modern ethics otherwise largely attempts but may struggle logically to reach and defend (including the protection of pluralism and multiculturalism which modern ethics (according to diverse critics including among its own champions) largely truncates or excludes).

The above references detail the extended, detailed, and more comprehensive defense of PSC compared to competing frameworks (especially the extended textbooks with Monlezun 2020 and Monlezun 2022). Furthermore, it is beyond the conceptual scope and space constraints of this work to decisively argue whether PSC is ultimately superior as a framework compared to others (the below description thus is only meant to demonstrate why the PSC is a suitable and sufficient framework for the specific task). Additionally, further definition and defense for the PSC were considered superfluous for this manuscript and irrelevant for the vast majority of readers given the largely uncontroversial and generally accepted ethical principles (including human rights) and the conclusions logically following from them. However, even the clinical and economic analyses in AiCE, if considered independent of the ethical dimension, are considered by this work to be sufficient reasoning to support collective action across diverse belief systems, healthcare systems, and states to improve the clinical challenge addressed in this work. Thus, the particular ethical framework diverse readers invoke explicitly or implicitly to reach this conclusion is beyond the scope of this paper. Its primary ethical framework is at least compatible with the vast majority of readers’ diverse ethical frameworks (a generally accepted claim) and at most is more compellingly argued using the paper’s PSC framework (a less generally accepted claim that still does not need to be proven in the brief confines of this work for the end of the conclusion to still hold, regardless of the particular means that diverse readers may take to arrive at it).

The core structural features of its framework are briefly described as follows. Metaphysically, it incorporates a Carlo-refined Clark-style Strong Thomistic Personalism that recognizes the person in her/his objective and subjective dimensions as a being who is most complete, happy, and flourishing in a gift-of-self specifically to other persons in love (the fullness of person-directed justice), and to other beings more generally in responsible care for the larger non-person ecosystem. As such, it entails an extended defense of a metaphysics of multiculturalism that explicitly cites the world’s diverse belief systems (including in their canonical texts as applicable) and elaborates the substantive converging (not simply Rawlsian-like overlapping) consensus as the metaphysical (not simply political) identity of the person individually and thus the criteria for justice and its subsequent peace communally in the community of persons globally. This consensus is a three-dimensional conception of human dignity that is logically derivative from the metaphysical identity of the human person grounded in the good as initially described classically by the physician-philosopher Aristotle. In the three metaphysical and thus personal dimensions of existential origin, moral order, and goodness orientation, the person can be understood (commonly across belief systems and through sufficiently respectful and careful exploration of those belief systems) to have intrinsic and non-finite (or arbitrarily limited) value. Justice, therefore, is giving to each their due, and to persons what is the only proper response to the unique individual human in front of us—the gift of self to the other self, proportional to the type of relationship between the person and the concrete context of that relationship. Logically and experientially derivative from this metaphysical foundation is the PSC’s theoretical principles (definition of and thus respect for individual dignity and communal culture (the latter being the collective and relational search for the ultimate good or goodness itself as the most fundamental, human, and personal of all endeavors and acts)). Its practical principles include solidarity and subsidiarity. Its primary ethical principle is the Wojtylan Personalist Norm (as a modification of Kant’s second categorical imperative, elevating the constructivist and minimalist Enlightenment ethical principle to the personal dimension, by arguing for love as the essence of a full conception of ethics, based on justice or what is due to persons from other persons, since the “person is a good towards which the only proper and adequate attitude is love”). These principles are relationally ordered in the pluralistic framework emerging from the above Thomistic Personalist metaphysical foundation by incorporating the unique perspectives in their own words of the world’s diverse belief systems (including Buddhism, Christianity, Confucianism, Daoism, Hinduism, Islam, Judaism, and non-religiously affiliated secularism (with particular attention paid to the nuances and subtleties among and between these religious frameworks including atheism and agnosticism)) [62,63,64,65,66,67,68,69].

In summary, the PSC argues that the world’s diverse belief systems converge existentially and substantively as well as metaphysically and ethically in the shared conviction of the intrinsic and inviolable dignity of every human person. This dignity is derivative from her/his biological identity as a human being (regardless of any artificially or arbitrarily identified traits such as sex, nationality, or belief system). As such, the person is a dependent rational animal from the earliest to the final moment of existence, linked in societal inter-dependencies requiring and fostering virtuous and thus just treatment to all members of the human community to survive and thrive. The community in turn is required for the full flourishing of the human person who finds her/his fulfillment (union with good itself) in the duty of justice contributing to the common good of the community, which in turn safeguards the individual good of the person (completed metaphysically in the highest form of justice, which is love, the commitment of the will to the objective good of the other person as other). The PSC defines and defends such convergence, which is individually echoed and anchored in the above diverse belief systems’ principles (with Buddhism’s *sila*, Christianity’s doctrine of Jesus’ incarnation and redemptive passion and resurrection, Confucianism’s *jen* and *yi*, Hinduism’s *dharma*, Islam and Judaism’s (along with Christianity’s) doctrine of humanity made in the image and likeness of God and destined for unity with God through a just life of love, and secularism’s Rawlsian-like political and pluralistic ‘justice’ as fairness).

### 2.12. Quality Control, Result Reporting, and Analytic Software

An academic physician-data scientist, biostatistician, and ethicist (DJM) confirmed that the final analytic models were sufficiently supported by the existing literature and related theories. Mean values are reported with standard deviation (SDs). Fully adjusted regression results were reported with 95% confidence intervals (CIs) with statistical significance set at a 2-tailed *p*-value of <0.05. Statistical analysis was performed with STATA 17.0 MP edition (STATACorp, College Station, TX, USA), and ML and DL analyses were performed with Java 9 (Oracle, Redwood Chores, CA, USA).

## 3. Results

### 3.1. Sample Descriptive Statistics and Bivariable Analysis by Cardiac Arrest

Of the 101,521,656 hospitalizations from 2016 to 2018 across 4550 hospitals nationally, 6,656,883 (6.56%) suffered cardiac arrest with a 56.10% mortality, LOS of 8.58 days (SD 14.14), mean cost of $159,768.40 (SD 270,321.90), and of whom 61,300 (0.92%) had active cancer (Table 1). Among patients with cardiac arrest, the prevalence of active cancer remained mostly stable from 2016 (19,280 (0.92%)) to 2017 (20,370 (0.90%)) to 2018 (21,650 (0.96%)). Across all years, on average, the presence versus absence of active cancer was significantly more likely in patients with cardiac arrest (0.92% versus 0.74%) and metastatic malignancy (1.04% versus 0.74%) (all *p* < 0.001). Patients with versus without active cancer were significantly less likely to receive an inpatient LHC (7.42% versus 20.79%) and more likely to die inpatient at mostly stable levels across all three years, though there was a stepwise increase in LHC among patients with cancer up to 7.74% by 2018 and decrease in mortality to 72.46% by 2018 (all *p* < 0.001) (Figure 1).

Among patients with cardiac arrest, those with versus without active cancer were significantly more likely to be African American (20.44 versus 18.60) and less likely to be in the lowest income quartile (31.61% versus 33.61%), have Medicaid (11.16% versus 13.81%), be uninsured (2.54% versus 4.39%), and live in a central metro region of ≤1 million people (67.43% versus 69.47%) (all *p* < 0.001). The regions with the highest prevalence of active cancer among patients with cardiac arrest included the South Atlantic (23.02%), Pacific (14.50%), East North Central (14.46%), Mid Atlantic (13.60%), and West South Central (12.25%) (*p* = 0.002).

In bivariable analysis among patients with cardiac arrest, patients with versus without active cancer were significantly less likely to receive an inpatient LHC (8.56% versus 20.43%, *p* < 0.001) and more likely to die inpatient (74.21% versus 54.91%, *p* < 0.001), which was comparable when matched by age and NIS-calculated mortality risk by DRG (72.31% versus 59.66%, *p* < 0.001).

### 3.2. Cardiac Catheterization and Mortality Disparities

According to the latest available U.S. census data (only reported up to one decimal point), the population distribution is non-Hispanic Caucasian (60.1%), Hispanic (18.5%), African American (13.4%), Asian (5.9%), Native American (1.3%), and other (0.8%) [70]. Within the 2016 NIS among patients with active cancer, the racial distribution of cardiac arrest included non-Hispanic Caucasian (63.2%), Hispanic (8.56%), African American (20.94%), Asian (3.43%), Native American (0.51%), and other (3.4%), leaving disparities by race in cardiac arrest prevalence versus population distribution among non-Hispanic Caucasians (+3.1%), Hispanics (−9.9%), African Americans (+7.5%), Asian (−2.5%), Native Americans (−0.8%), and others (+2.6%).

Within the 2016 NIS among patients with active cancer, the racial distribution of LHC following cardiac arrest included non-Hispanic Caucasian (78.9%), Hispanic (2.7%), African American (12.6%), Asian (1.9%), Native American (0.8%), and other (3.1%), leaving disparities in LHC prevalence by race versus population distribution among non-Hispanic Caucasians (+18.8%), Hispanics (−15.8%), African Americans (−0.8%), Asians (−4.0%), Native Americans (−0.5%), and others (+2.3%).

Within the 2016 NIS among patients with active cancer, the racial distribution of mortality following cardiac arrest included non-Hispanic Caucasian (61.0%), Hispanic (9.3%), African American (21.8%), Asian (3.75%), Native American (0.5%), and other (3.6%), leaving disparities in mortality by race versus population distribution among non-Hispanic Caucasians (+0.9%), Hispanics (−9.2%), African Americans (+7.8%), Asians (−2.1%), Native Americans (−0.8%), and others (+2.8%).

LHC disparities (age and risk matched) in cancer versus non-cancer were more pronounced for males (8.88% versus 20.90%, *p* < 0.001) than females (2.99% versus 14.35%, *p* = 0.009), Caucasians (10.01% versus 22.16%, *p* < 0.001) than non-Caucasians (4.90% versus 15.36%, *p* < 0.001), highest income quartiles (9.62% versus 21.55%, *p* < 0.001) than the lowest (6.27% versus 17.52%, *p* < 0.001), and urban metros of at least 1 million residents (7.29% versus 20.28%, *p* < 0.001) than fewer (10.10% versus 20.58%, *p* < 0.0001).

Matched by age and risk, mortality disparities for patients with versus without active cancer were more pronounced for females (79.10% versus 63.45%, *p* = 0.010) than males (71.98% versus 59.35%, *p* < 0.001, non-Caucasians (77.20% versus 62.29%, *p* < 0.001) than Caucasians (70.40% versus 58.08%, *p* < 0.001) than, highest income quartiles (71.80% versus 58.83%, *p* < 0.001) than the lowest (73.80% versus 61.51%, *p* < 0.001), and urban metros of at least 1 million residents (72.88% versus 60.24%, *p* < 0.001) than fewer (71.55% versus 58.99%, *p* < 0.0001).

### 3.3. Propensity Score Adjusted Multivariable Regression of Mortality by Cardiac Catheterization

In ML-PS multivariable regression among patients with cardiac arrest and active cancer from 2016 to 2018, adjustment was performed for age, sex, cancer (leukemia, pancreas, liver, biliary, and lung), metastasis, pulseless electrical activity (PEA), ST-segment myocardial infarction (STEMI), mortality risk (as calculated by the NIS using DRGs), and the likelihood of receiving LHC. LHC significantly reduced mortality (OR 0.18, 95%CI 0.14–0.24, *p* < 0.001). The marginal effect for LHC was −38.71% (95%CI −45.47–−31.95; *p* < 0.001). Subgroup analysis within each of the 26 primary malignancies indicated that the only malignancies that were significantly associated with independently increased mortality were leukemia, pancreas, liver, biliary, and lung (below referred to as “high-risk malignancy”).

### 3.4. Mortality by LHC in Propensity Score Matching

In propensity score matching with the same above adjustment variables, LHC significantly and independently reduced mortality by 42.87% (95%CI 35.56–50.18, *p* < 0.001).

### 3.5. Creation of the Clinical Predictive Model of Cardiac Arrest in Active Cancer: CACOS

Baseline traits of the derivation and verification cohorts were not significantly different. CACOS included the following predictors: no inpatient LHC (OR 4.74, 95%CI 3.15–7.13), PEA initial rhythm (OR 2.32, 95%CI 1.78–3.02), metastatic malignancy (OR 1.80, 95%CI 1.41–2.30), and high-risk malignancy (OR 1.60, 95%CI 1.27–2.02) (all *p* < 0.001). The model developed among patients with active cancer generated a receiver operating curve (ROC) area under the curve (AUC), sensitivity, specificity, and correctly classified of 0.685, 97.51%, 17.75%, and 78.22% for the derivative and 0.668, 97.92%, 19.96%, and 76.63% for the verification sub-datasets. In a univariate regression, each increasing CACOS point was associated with a significant increase in mortality compared to a score of zero in a progressive stepwise linear fashion: one point (OR 2.61, 95%CI 1.55–4.39), two points (OR 6.75, 95%CI 4.14–11.01), three points (OR 14.00, 95%CI 8.56–22.90), and four points (OR 18.47, 95%CI 10.78–31.63) (all *p* < 0.001) (Figure 2).

### 3.6. Calibration of CACOS Predictive Model

Calibration plots for CACOS in the 2016 NIS derivation and verification sub-samples demonstrated sufficient calibration in both sub-samples (Figure 3 and Figure 4).

### 3.7. Two Separate External Validations of the CACOS Predictive Model

Two separate external validations of CACOS were conducted using the 2017 and then the 2018 NIS datasets among patients with active cancer following cardiac arrest. In the 2017 NIS, CACOS among patients with active cancer post-arrest generated an AUC, sensitivity, specificity, and correctly classified of 0.634, 96.51%, 17.81%, and 75.46%. This resulted in a percentage change in discrimination (with ROC) of 0.51% versus the CACOS derivation. In the 2018 NIS, CACOS in post-patients with active cancer generated an AUC, sensitivity, specificity, and correctly classified of 0.643, 96.62%, 17.45%, and 74.82%. This resulted in a percentage change in discrimination of 0.42% versus the CACOS derivation.

### 3.8. Deep Learning versus ML-PSr Performance with CACOS

DL analysis was conducted ultimately with five hidden layers (consisting, respectively, of 5, 3, 2, 10, and 5 nodes after alternative combinations of hidden layers and nodes were confirmed to generate suboptimal performance). The model achieved a superior ROC compared to the above regression model in the above Section 3.3 (AUC 0.695) (Figure 5).

### 3.9. Cost-Effectiveness and Cost Benefit Analysis

The cost-effectiveness of LHC for patients with cardiac arrest and active cancer was $915.82 spent to avert one additional death. The net benefit for the above intervention was $59.72 billion.

If LHC prevalence and the related mortality reduction (from PSM above) were equally distributed across all races according to their U.S. census population distribution, then 250 additional Hispanic patients, 31 African Americans, and 11 Asians may have been saved with additional net savings of $1.85 billion, $0.23 billion, and $80.58 million for a total of 292 additional minorities and $2.16 billion saved.

### 3.10. AI-Driven Computational Ethical and Policy Analysis: Personalist Social Contract

The above health and economic results then informed the final or focused ethical-policy analysis step of AiCE. The primary material object of this ethical analysis was LHC, the primary context was inpatient healthcare delivered to patients with active cancer following sustained ROSC after cardiac arrest, and the primary formal object or ethical analytic framework is the PSC.

Applied to this concrete ethical situation, the formal PSC argument is as follows. (Premise 1) Cardiac arrest in active cancer diagnosed inpatient carries a high mortality, which may be significantly reduced with LHC. (Premise 2) There appear to be significant disparities among patients with active cancer in the prevalence of fatal out-of-hospital cardiac arrest as patients of particular racial minorities, lower income, Medicaid, and no insurance may be more likely to present to the hospital. After the presentation, they appear less likely to receive LHC as do patients overall with versus without active cancer. Such disparities within healthcare systems suggest reduced effective prevention and treatment for reasons at least in part societal and not solely attributed to clinical differences across patients. (Premise 3) Life and equal societal protection are fundamental individual and state rights logically derivative from the human person’s dignity and are politically enshrined across the United Nations, multiple other international institutions, and the majority of nations’ constitutions and legal statutes. (Premise 4) Respect for dignity at the individual level requires respecting the person’s rights to goods (beginning with the primary good of life) necessary for the person to develop through a just and stable commitment to the common good and thus the community in reciprocal care for the individual. (Premise 5) Respect for dignity at the communal level requires respecting other cultures as the communal manifestations of their constitutive individuals seeking through justice the common good (as the objective good of the community, entailing the objective good of individual flourishing, and subjectively experienced as the ultimate individual good of self-actualization through justice, completed in love, uniting the person to the community which is united and animated by goodness itself). (Premise 6) Social disparities (in cardiac arrest prevention and mortality-reducing treatment) including race, income, and insurance can produce disproportionate mortality in those social sub-communities resulting in a disproportionate threat to the preservation of those persona and related cultures, leading to the global human community’s impoverishment with the loss or diminishment of those individuals and cultures. (Premise 7) The reduction of such disparities may result in hundreds of lives saved along with billions of dollars. (Premise 8) Continued disparities in prevention and treatment of cardiac arrest in patients overall compared to patients with active cancer and within demographic and socioeconomic subgroups within patients with active cancer undermine respect for the rights of patients and respect for their cultures, which is critical to the wellbeing of societies that encompass all peoples and cultures. (Premise 9) The CACOS score, particularly when automated as a clinical decision support tool within diverse existing EHRs across healthcare systems using only admission ICD10s, may additionally allow early refinement of the accuracy and precision with which post-arrest treatment is provided by more sensitive risk stratification for those likely to survive and thus benefit from more intense treatment efforts. (Conclusion) Therefore, clinical, economic, and ethical justification supports greater healthcare policy and healthcare system investment reducing disparities in the burden of cardiac arrest including with the use of such early scoring systems as CACOS.

## 4. Discussion

This is the first known AI and causal inference statistical-informed clinical, cost, and ethical integrated analysis of cardio-oncology cardiac arrests including LHC-related mortality reduction and related disparities. It is additionally the first such study to generate and validate a novel clinical predictive model for cardiac arrest in active cancer. By using the most recent multi-year nationally representative dataset, this study suggests that patients with versus without active cancer are approximately one-third as likely to receive inpatient LHC (though they are receiving LHC at increasing rates concurrent with decreasing mortality over time) as LHC may reduce mortality by upwards of 42.87%. However, there appear to be significant disparities in cardiac arrest prevalence, treatment, and outcomes. Our results suggest among post-arrest patients that Hispanic, African American, lower income, Medicaid, uninsured, and more urban patients with active cancer may suffer from higher undiagnosed or under-managed chronic comorbidities. Additionally, their culmination in cardiac arrest may be disproportionately suffered by such populations given their lower inpatient diagnosis prevalence of cardiac arrest compared to their baseline population distribution (while post-arrest Hispanics and African Americans with active cancer are less likely to receive inpatient LHC and African Americans more likely to die while hospitalized). Given such disparities and this study’s cost-effectiveness and benefit analysis suggesting that less than $1000 needs to be spent on average on an inpatient LHC to save the life of a patient with cardiac arrest and active cancer, our analysis suggests that nearly 300 racial minorities and over $2 billion annually are lost nationally because of persistent disparities in cardiac arrest inpatient treatment. Computational ethical analysis indicates that across our diverse nations, healthcare systems, and belief systems (whether Buddhism, Christianity, Confucianism, Daoism, Hinduism, Islam, Judaism, or secularism), there is robust empirical and ethical evidence that such disparities should be reduced for the equitable benefit for cardio-oncology patients globally for a net societal benefit. Our novel CACOS clinical predictive model for post-cardiac arrest mortality in patients with active cancer may provide a concrete step in this direction by early sensitive risk stratification of patients to allow healthcare systems and nations (particularly those with fewer resources) the ability to more precisely match those limited resources for patients who may most benefit from aggressive post-arrest care.

Prior research using a novel integration of AI, propensity score analysis, and geographic information system (GIS) heat mapping has demonstrated that lower-income racial minorities (at an individual and community level) not only suffer higher actual rates of cardiac arrest but also post-arrest poor neurological outcomes [71]. By accurately showing the geographic overlap at the neighborhood level of cardiac arrest burden and residence of lower-income racial minorities, such research can help inform more culturally sensitive and effective local healthcare system and health policy outreach efforts to match resources with where their demand and the potential benefit is the greatest (to treat the patient and her/his community concurrently through the complementary collaboration of clinical medicine and public health). Such approaches with innovative methodologies and proposed applications may be helpful in reducing the largely static high burden (and related disparities) of cardiac arrest in the last few decades. Prior research has demonstrated that such disparities persist in pre-arrest risk factors, cardiac arrest incidence, CPR, guideline-recommended treatment, palliative care (including lower ICD implantation among women and racial minorities regardless of hospital traits and comorbidities), and adjusted survival to hospital discharge [72,73,74,75,76]. Notably, Starks et al. 2017 demonstrated that compared to predominantly Caucasian neighborhoods, patients from predominantly African American neighborhoods are less likely to receive post-arrest bystander CPR and defibrillation and thus survive ultimately to hospital discharge, with African American communities having 38.30% lower bystander CPR rates than Caucasian communities. Further, large recent trials including in the *New England Journal of Medicine* increasingly suggest that cardiac catheterization can be delayed for patients with cardiac arrest with admission non-STEMIs in general, and thus there has been a growing trend of delayed LHC even in the outpatient setting [77,78]. This study provides novel evidence that patients with active cancer may particularly benefit from inpatient LHC.

Such cardiac arrest disparities from the general population are reasonably expected to hold in the sub-group of patients with active cancer, and yet the limited research has further limited effective action on the already daunting challenge of persistent disparities. This study thus not only helps to accurately describe the epidemiology nationally of cardiac arrest in active cancer but also does so with a granular assessment by primary malignancy and related disparities by sex, income, insurance, and region (by arrest prevalence, treatment, and outcomes). This study also creates and validates the novel tool of CACOS which may accurately predict post-arrest mortality with commonly available variables usually at the time of admission (which may allow enhanced clinical decision support to optimize health outcomes and health equity through more precise matching of resources to not only patient need but also their likelihood to benefit from it). The AI (a) analytics and (b) applications for this study and particularly CACOS additionally suggest a paradigm pivot in such clinical research to accelerate value-based healthcare performance by healthcare systems. The above results were demonstrated using a large nationally representative dataset and robust causal inference statistics including the propensity score adjustment and matching in a way that is generally well accepted by the diverse clinician and researcher audiences. It also confirms these results using (a) AI analytics (both ML and DL). Growing research emblematically demonstrated by *Nature*’s Popescu et al. 2022 shows how AI is increasingly moving modern medicine from cruder population-based predictions to more precise patient-specific predictions for a new era of enhanced outcomes. Such AI-augmented analyses, such as our study demonstrates, suggest how AI can achieve at least comparable results to the well-accepted traditional statistical techniques dominating contemporary clinical research. However, unlike traditional statistics, it has the increasingly operationalized potential to achieve new levels of accuracy and precision in real-time using the unprecedented Bid Data volume, velocity, and variety generated by modern healthcare systems [79]. Additionally, CACOS is meant to have innovative (b) AI applications unlike prior predictive scores and traditional statistics—by placing CACOS within diverse healthcare systems’ EHRs (that can rapidly pull ICD10s) it can provide real-time guidance for clinicians to improve related patients’ equitable outcomes while allowing its underlying AI algorithms to continually adapt the model and personalize it for the unique patient.

This AiCE analysis and intended application suggest the importance of AI augmentation with the co-design (collaboratively with patients, community members, clinicians, data scientists, policymakers, and healthcare system executives) of ethically aligned AI to build transparency, reliability, and trust for diverse stakeholders within and outside the healthcare system. This hybrid approach of traditional statistics and AI analytics may thus be instrumental by additionally serving as a methodological bridge from older statistics to next-generation AI. This new paradigm proposes a potentially promising model of the future’s next-generation AI-empowered, health equity-focused, value-based healthcare-orientated healthcare system [80].

This study, therefore, operates within this conceptual model of Health AI which can be formally defined as the emerging model of the AI-driven or ‘thinking healthcare system’ and mathematically defined as the following (Monlezun 2022):$$\mathrm{Health}\;\mathrm{AI}={\left(\mathrm{HealthBD}\times \left[\bar{\mathrm{Delivery}}+\overset{\infty }{\underset{n=1}{\sum }}\left\{\mathrm{PrMed}\;\langle \cos\mathrm{Delivery}\rangle +\;\mathrm{PubHealth}\;\langle \sin\mathrm{Delivery}\rangle \right\}\right]\right)}^{\mathrm{AI}-\mathrm{VBHC}}$$

Conceptually, AI Health is the product of Healthcare Big Data (HealthBD) and healthcare delivery, raised to the power of AI-enabled Value-Based Healthcare (AI-VBHC). If Health AI is meant to conceptualize the primary objective of the emerging model of healthcare systems, the means to it is the AiCE (utilized in this study and) central to its AI-VBHC transformation. AI-VBHC specifically can be formulaically represented as:$$\mathrm{AI}\;\mathrm{VBHC}=\mathrm{AiCE}\times \left({\left[\mathrm{Clinical}+\mathrm{Operational}\right]}_{\mathrm{AI}}\right)\times \left(\frac{{\mathrm{Quality}}_{\mathrm{Equity}\;\times \;\mathrm{Personal}\;\times \;\mathrm{Social}\;\times \;\mathrm{Wellbeing}}\;}{{\mathrm{Cost}}_{\mathrm{Time}\;\times \;\mathrm{Capacity}\;\times \;\mathrm{Support}}}\right)$$

Healthcare delivery as a periodic function in Health AI is represented as the trigonometric form of a Fourier series as the infinite convergent series of the sum of the average unit of healthcare delivery at a patient level (as the average value of the function) and the summation of the cosine wave (of PrMed or Personalized Medicine) and summation of the sine wave (of PubHealth or Public Health) [81]. The successive sum of these waves or harmonics (integer multiples of the periodic function’s fundamental frequency) constituting the overall PrMed and PubHealth waves allows the convergence or increasing approach toward the limit as the function or number of terms increases (as the function *y* = 1/*x* converges to zero as *x* increases). This healthcare delivery framework represents the operational function of healthcare systems whose central objective should be to deliver VBHC, whose seminal definition was provided by Porter and Teisberg as a cost-benefit function of health outcomes achieved through quality healthcare divided by the per-person costs to achieve those outcomes [82,83]. The European Commission’s Expert Panel on Effective Ways in Investing in Health refined the definition of quality healthcare as consisting of equity, person-centeredness, social participation, and wellbeing with implied safety [84]. The AI-driven Cardiac Arrest Inequity Index (AI-CAII) and the AI-driven Efficiency-Inequity Index (AI-EII) are derivative of the above as the former is the ratio of observed over predicted outcomes applied to cardiac arrest. The latter describes the more general clinical and technical trade-off in efficiency reaching the desired value-based healthcare outcomes, and the disparities that may increase with the efficiency jumps (thus guiding optimal efficiency boosts without disproportionate disparity increases). The AI-CAII for this study highlights the potential preventable mortality and costs from arrest disparities, while the AI-EII remains appropriately in the desired range over 1 as the efficiency of CACOS deployment in EHRs (and non-EHRs for lower resource communities) reasonably outpaces any disparity it may produce.

This study is consistent with prior research suggesting the growing number and application of healthcare digital innovations related to the Fourth Industrial Revolution (Industry 4.0) [29]. Similar to prior industrial revolutions which introduced commercialized scientific-techno innovations with subsequent force multiplying transformation of other economic sectors (and societies more broadly), Industry 4.0 emerged in the 2010s from the Third (Digital) Industrial Revolution marking the fundamental global shift to embedded intelligent connectivity of cyber-physical systems with our augmented social reality (driven by AI-enabled smart technologies including in the Internet of Things [IoT] and advanced robotics). Similar to other economic sectors, Industry 4.0 is rapidly facilitating the AI-accelerated horizontal (external) and vertical (internal) digital integration of value-based supply chains producing a digital ecosystem that connects healthcare systems with more dense and digitized networks of telehealth vendors, pharmaceuticals, medical device companies, insurance companies, technological companies, community organizations, governments, and non-governmental organizations. Internally, healthcare systems are increasingly adopting the Industry 4.0 hierarchical information technology (IT) architecture of cloud layer (internet-based servers for data storage, processing, analytics, and related user services) and their underlying fog layer (intermediary servers amplifying and accelerating data processing, caching, buffering, and communication) and most remote edge layer (where users, i.e., patients interface with the digital counterpart of the physical system through IoT devices such as smartphones and remote sensors which generate data and access architecture’s services such as telehealth). This unprecedented volume, velocity, and variety of this increasingly dense Industry 4.0 (including healthcare) data are indicated as Big Data which AI is increasingly required to process, analyze, and communicate with the different components of the digital ecosystem informing clinical and operational decisions for healthcare systems.

Accordingly, there are a growing number of AI-enabled Big Data innovations including using the IoT that are improving the effectiveness, efficiency, and equity of traditional healthcare operations [85]. In public health, deep extreme learning has been used to achieve an accuracy rate of 97.59% forecasting COVID-19’s spread [86]. In precision medicine, Adel et al. 2021 demonstrated that clinicians can more rapidly extract needed patient data from EHRs using a fuzzy ontology-based semantic interoperability framework with an ML-based natural language processing (NLP) approach [87]. There has been extensive work in AI-augmented or intelligent imaging to improve monitoring of fetal organ development with ultrasounds (using DL with convolutional neural network (CNN)), COVID diagnoses with chest X-rays (using CNN), and even dental age in forensics (using neural networks and X-rays) [88,89,90]. As such AI applications continue, there are concurrent efforts, including the blockchain (decentralized digital ledger distributed across a network of peer-to-peer servers), to preserve the security, integrity, and privacy of such sensitive data from internal accidental information loss and external deliberate cyberattacks [91]. The above results should be cautiously interpreted given the study’s limitations. This is a non-randomized observational study using only U.S. administrative de-identified data (not able to longitudinally track at the individual level) which were used to generate a clinical predictive model with a lower specificity and AUC than typically more general (not specifically concurrent active cancer in) cardiac arrest models. This study sought to reduce the threats to internal and external validity by utilizing a large nationally representative dataset from three separate years, robust traditional causal inference statistics (including propensity score adjustment and matching), both ML and DL, and repeated validations of the predictive model. The design decision to optimize CACOS sensitivity even at the expense of specificity and overall AUC was considered of sufficient net benefit, considering its still sufficient AUC for such a novel cancer-focused arrest predictive model and its objective to enhance its clinical utility for rapid risk stratification for such patients near the time of initial admission (when much of the projected hospital-based treatment plan is initially formulated).

## 5. Conclusions

This is the first known AI and causal inference statistical-informed clinical, cost, and ethical integrated analysis of cardio-oncology cardiac arrests including LHC-related mortality reduction and related disparities, in turn, used to generate and validate the first known clinical predictive model for cardiac arrest in active cancer (CACOS). By using the most recent multi-year nationally representative dataset, this study suggests that patients with versus without active cancer are less likely to receive inpatient LHC (though they are receiving LHC at increasing rates concurrent with decreasing mortality over time). This disparity is particularly concerning given how LHC may reduce mortality by upwards of 42.87%. However, there appear to be significant disparities in cardiac arrest prevalence, treatment, and outcomes including by race, income, insurance, and region that empirical and ethical evidence justifies reducing as quickly as possible. As such, this methodological approach of AiCE within AI Health (with its above mathematical definition) proposes not a rigid imposing roadmap but rather an inclusive formula for the future of healthcare systems that are effective, efficient, and equitable (by being automatable and adaptable for local communities’ needs and values, rather than being dictated externally).

## Figures and Tables

**Figure 1 medicina-58-01039-f001:**
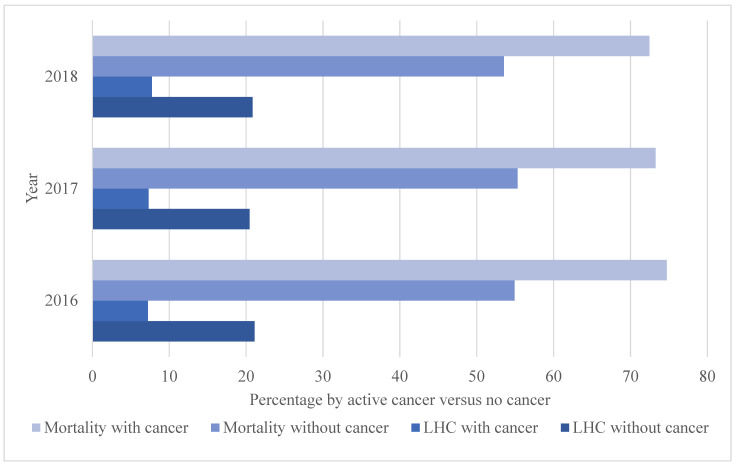
Longitudinal analysis of inpatient mortality and left heart catheterization (LHC) in cardiac arrest by active cancer (*N* = 6,656,883).

**Figure 2 medicina-58-01039-f002:**
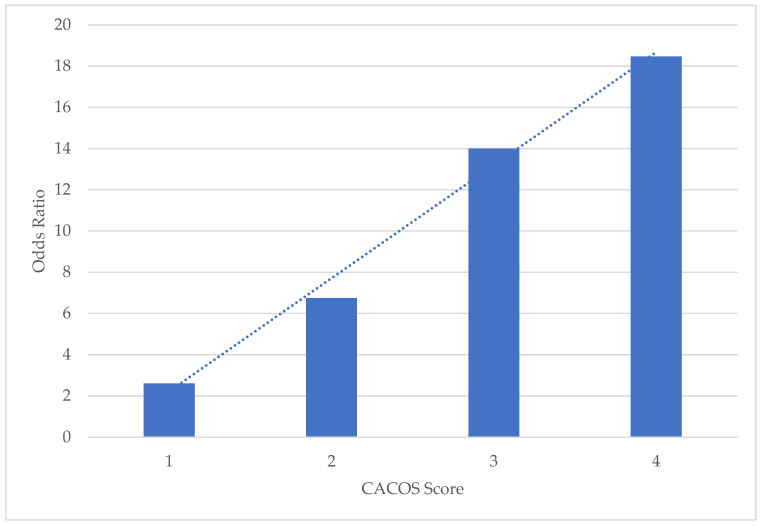
Inpatient mortality post-cardiac arrest in active cancer by Cardiac Arrest Cardio-Oncology Score (CACOS) according to univariable regression (*N* = 19,280).

**Figure 3 medicina-58-01039-f003:**
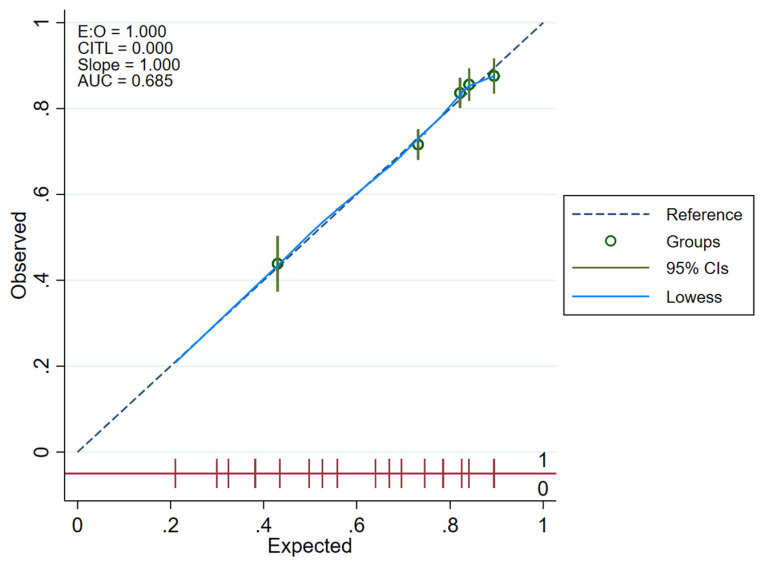
Calibration plot for Cardiac Arrest Cardio-Oncology Score (CACOS) for the derivation sub-sample.

**Figure 4 medicina-58-01039-f004:**
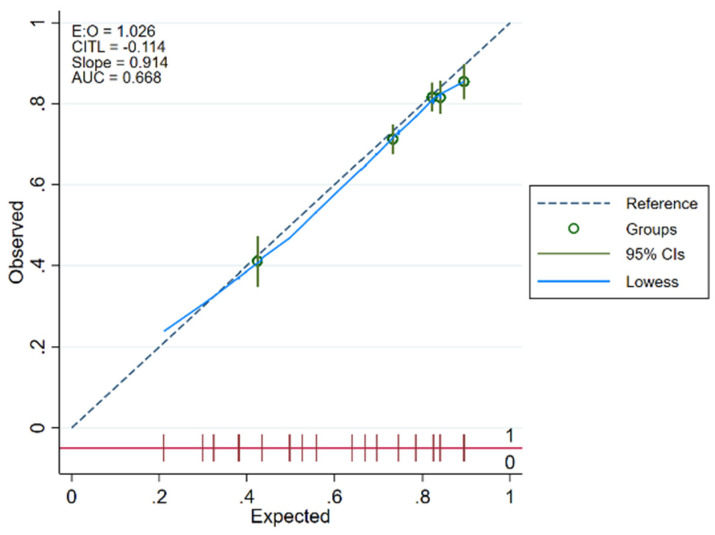
Calibration plot for Cardiac Arrest Cardio-Oncology Score (CACOS) for the verification sub-sample.

**Figure 5 medicina-58-01039-f005:**
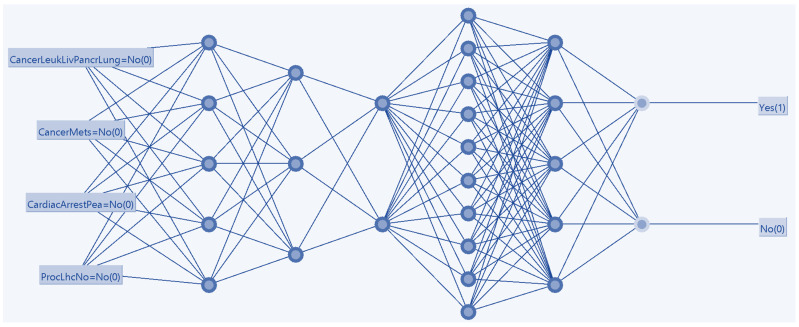
Deep learning model with a backward propagation artificial neural network of cardiac arrest morality in active cancer (*N* = 6,656,883).

**Table 1 medicina-58-01039-t001:** Descriptive and bivariable analysis by active cancer in cardiac arrest by year (*N* = 6,656,883).

Variables, %	2016–2018				2016				2017				2018			
	Sample	No-Cancer	Cancer	*p*-Value	Sample	No-Cancer	Cancer	*p*-Value	Sample	No-Cancer	Cancer	*p*-Value	Sample	No-Cancer	Cancer	*p*-Value
	6,656,883	6,595,583	61,300		2,145,699	2,126,419	19,280		2,253,189	2,232,819	20,370		2,257,995	2,236,345	21,650	
Demographics																
Age, mean (SD)	63.66 (16.99)	64.41 (17.26)	67.48 (13.22)	<0.001	65.33 (15.80)	65.14 (16.02)	67.55 (12.90)	<0.001	61.19 (17.68)	63.91 (17.98)	67.43 (13.19)	<0.001	64.45 (17.50)	64.18 (17.78)	67.46 (13.57)	<0.001
Female	41.08	41.10	40.81	0.311	41.36	41.49	39.91	0.056	41.02	41.04	40.72	0.689	40.86	40.77	41.80	0.188
Race				<0.001				<0.001				<0.001				<0.001
White	65.32	65.49	63.45		66.29	66.57	63.17		65.25	65.27	64.95		64.43	64.63	62.24	
Black	18.75	18.60	20.44		18.60	18.39	20.94		18.85	18.82	19.18		18.81	18.59	21.20	
Hispanic	9.33	9.37	8.96		8.66	8.67	8.56		9.34	9.39	8.80		10.00	10.04	9.52	
Asian	2.84	2.78	3.57		2.84	2.79	3.43		2.79	2.69	3.89		2.90	2.86	3.38	
Native American	0.62	0.62	0.51		0.58	0.58	0.51		0.65	0.66	0.46		0.62	0.62	0.57	
Income				0.002				0.005				0.001				0.001
1st (lowest)	33.45	33.61	31.61		33.92	34.06	32.34		33.85	34.05	31.52		32.58	32.72	30.97	
2nd	26.34	26.38	25.89		25.36	25.41	24.72		26.54	26.54	26.53		27.13	27.19	26.42	
3rd	22.56	22.54	22.83		22.86	22.84	23.11		22.29	22.25	22.76		22.53	22.52	22.63	
4th (highest)	17.65	17.47	19.66		17.87	17.69	19.83		17.32	17.16	19.19		17.77	17.57	19.97	
Insurance				<0.001				0.001				<0.001				<0.001
Commercial	19.65	19.45	21.84		19.91	19.70	22.20		19.70	19.53	21.71		19.34	19.13	21.62	
Medicare	59.59	59.42	61.46		60.05	59.96	61.02		59.15	58.94	61.54		59.56	59.35	61.83	
Medicaid	13.60	13.81	11.16		12.98	13.15	11.14		13.93	14.17	11.09		13.88	14.11	11.25	
VA	2.93	2.93	3.00		2.94	2.93	3.02		2.98	2.97	3.17		2.88	2.89	2.80	
None	4.24	4.39	2.54		4.12	4.26	2.63		4.25	4.39	2.48		4.34	4.51	2.50	
Urban				0.021				0.060				0.004				<0.001
≥1 million central	30.69	30.53	32.58		30.16	30.11	30.78		30.63	30.43	33.00		31.28	31.04	33.95	
≥1 million fringe	22.76	22.69	23.63		22.81	22.67	24.37		22.91	22.91	22.95		22.57	22.48	23.58	
250,000–999,999	21.21	21.33	19.81		21.34	21.47	19.86		21.24	21.29	20.64		21.05	21.24	18.94	
50,000–249,999	9.17	9.25	8.30		9.17	9.22	8.65		9.15	9.26	7.86		9.20	9.28	8.38	
Micro	9.34	9.36	9.12		9.47	9.48	9.36		9.24	9.26	9.01		9.30	9.33	8.98	
<Micro	6.82	6.84	6.57		7.05	7.06	6.98		6.82	6.84	6.55		6.59	6.63	6.17	
Region				0.002				<0.001				0.005				<0.001
New England	3.71	3.70	3.84		3.76	3.74	4.02		3.70	3.69	3.83		3.67	3.67	3.67	
Mid Atlantic	11.85	11.70	13.60		12.14	11.98	13.90		11.83	11.67	13.75		11.58	11.44	13.16	
East North Central	15.59	15.31	14.46		15.55	15.66	14.29		15.47	15.54	14.63		15.74	14.73		
West North Central	5.93	5.96	5.66		5.70	5.74	5.26		6.01	6.00	6.14		6.08	6.13	5.59	
South Atlantic	22.66	22.63	23.02		22.63	22.58	23.18		22.82	22.79	23.15		22.54	22.52	22.73	
East South Central	7.69	7.75	7.04		7.82	7.93	6.56		7.69	7.74	7.09		7.56	7.57	7.46	
West South Central	12.68	12.71	12.25		12.63	12.60	12.97		12.63	12.66	12.27		12.77	12.88	11.52	
Mountain	6.09	6.14	5.53		6.13	6.18	5.55		6.11	6.17	5.47		6.02	6.06	5.57	
Pacific	13.83	13.77	14.50		13.64	13.59	14.26		13.73	13.74	13.67		14.12	13.99	15.57	
Past medical history																
Cancer																
Active	0.75	0.74	0.92	<0.001	0.79	0.78	0.90	<0.001	0.73	0.72	0.90	<0.001	0.74	0.73	0.96	<0.001
Prior	2.94	3.15	0.48	<0.001	0.79	0.80	0.71	<0.001	7.30	7.92	0.00	<0.001	0.74	0.74	0.73	0.230
Metastatic	0.75	0.74	1.04	<0.001	0.79	0.78	1.00	<0.001	0.73	0.72	1.03	<0.001	0.74	0.73	1.10	<0.001
HTN	59.21	59.53	52.21	<0.001	67.80	68.37	61.28	<0.001	55.19	55.37	53.14	0.006	54.63	54.84	42.22	0.001
CAD	39.02	40.19	25.55	<0.001	40.21	41.41	26.53	<0.001	38.41	39.56	24.79	<0.001	38.43	39.60	25.33	<0.001
Afib	29.54	29.81	26.47	<0.0001	29.35	29.65	25.96	<0.001	29.18	29.48	25.63	<0.001	30.08	30.29	27.81	0.001
CHF	19.37	19.95	12.84	<0.0001	28.49	29.44	17.63	<0.001	13.92	14.26	9.94	<0.001	15.71	16.14	10.95	<0.001
Cirrhosis	3.44	3.36	4.35	0.001	3.31	3.22	4.33	<0.001	3.48	3.41	4.30	0.003	3.52	3.44	4.41	0.001
CKD 3–5	22.69	23.22	16.61	<0.0001	22.29	22.87	15.72	<0.001	22.65	23.19	16.18	<0.001	23.13	23.59	17.94	<0.001
Prior MI	9.37	9.63	6.40	<0.0001	9.75	10.06	6.28	<0.001	9.06	9.31	6.19	<0.001	9.30	9.53	6.72	<0.001
Prior treatment																
Chemotherapy	-	-	8.78	-	-	-	8.74	-	-	-	8.76	-	-	-	8.85	-
Radiation	-	-	5.12	-	-	-	4.59	-	-	-	5.33	-	-	-	5.43	-
Immunotherapy	-	-	0.03	-	-	-	0.03	-	-	-	0.05	-	-	-	0.00	-
Inpatient intervention																
LHC	19.72	20.79	7.42	<0.001	19.97	21.09	7.21	<0.001	19.42	20.45	7.31	<0.001	19.76	20.84	7.74	<0.001
PCI	13.35	14.14	4.36	<0.001	20.57	21.73	7.37	<0.001	9.69	10.29	2.58	<0.001	9.79	10.39	3.12	<0.001
Outcomes																
Mortality	56.10	54.58	73.47	<0.001	56.51	54.91	74.71	<0.001	56.71	55.31	73.25	<0.001	55.09	53.53	72.46	<0.001
LOS, mean days (SD)	8.58 (14.14)	8.56 (14.13)	8.79 (13.85)	0.409	8.45 (13.33)	8.42 (13.35)	8.78 (13.08)	0.116	8.66 (14.57)	8.64 (14.50)	8.92 (15.35)	0.238	8.63 (14.51)	8.63 (14.63)	8.67 (13.12)	0.872
Cost, mean ($)	159,768.40 (270,321.90)	161,014.23 (271,884.73)	145,542.57 (251,257.27)	0.001	145,808.80 (225,198.40)	147,010.80 (227,305.90)	132,107.00 (199,122.70)	<0.001	163,057.90 (286,292.00)	164,198.10 (287,488.80)	149,624.20 (271,479.80)	0.002	170,438.50 (299,475.30)	171,833.80 (300,859.50)	154,896.50 (283,169.30)	<0.001

SD, standard deviation; VA, Veterans’ Affairs; Mid, Middle; HTN, hypertension; CHF, congestive heart failure; afib, atrial fibrillation; COPD, chronic obstructive pulmonary disease; CKD 3–5, chronic kidney disease stages 3–5; MI, myocardial infarction; CVA, cerebrovascular accident; LHC, left heart catheterization; PCI, percutaneous coronary intervention; LOS, length of stay; $, United States dollars.

## Data Availability

The data used to support the findings of this study are publicly available from the cited sources.

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
