# Peer review of "Artificial Intelligence-Augmented Propensity Score, Cost Effectiveness and Computational Ethical Analysis of Cardiac Arrest and Active Cancer with Novel Mortality Predictive Score"

_medicina, 2022, doi:10.3390/medicina58081039_

Round 1

Reviewer 1 Report

Dear authors,

Thank you for making good effort in adding knowledge to the existing. I have reviewed your paper and find it sound enough to be published in this journal. However, my only concern is that the title is somewhat too long with alot of key words. This makes it complex in uderstanding the major focus of the paper. Do well to summarize the title to better focus on the core intent of the research.

Author Response

Thank you for the helpful recommendations. Our team sincerely hopes that the attached revised file satisfactorily addresses them to justify publication (with point-by-point response in CAPS below to your specific recommendations and tracked changes included in the attached file).

  1. However, my only concern is that the title is somewhat too long with alot of key words. REVISED (IN ADDITION TO THE REVISIONS REQUESTED BY THE OTHER REVIEWER).

Reviewer 2 Report

In this paper, a nationally representative cohort analysis was performed for mortality and cost by active cancer using the largest United States all-payer inpatient dataset. The structure of this paper is relative well organized, however the major revisions are needed for improvement. The comments are listed below.

1. This paper is a case study for Artificial Intelligence-augmented analysis. For the challenges, have authors investigated the solutions, this point can be clearer in the revision.

2. The discussions on other areas how people deal with the same issues can be covered.

3. For the evaluation, the dataset/algorithm choosing reasons, the detailed platform configurations and the discussion on other untested datasets should be introduced in the revised paper.

4. Some abbreviations (e.g., AI) are not explained where they first appear.

5. Please add a notation table to explain all used math symbols for easy understanding.

6. Please go through the paper carefully and double check whether the right template are used. Correct some errors and formatting issues (e.g., “ and nations)[7,8].” -> “ and nations) [7,8].” ? The necessary space is likely missing).

7. Some references lack the necessary information (e.g., [9]), please provide all information according to the right template.

8. Resize some oversized figures (e.g., Figure 2).

9. The writing of this paper must be well improved, some sentences are too long to understand.

10. Make the References more comprehensive, besides Artificial Intelligence-augmented analysis, some other promising scenarios (e.g., Big data, other IoT systems) can be covered in this work. If the above related work can be discussed, it can strongly improve the research significance. For the improvement, the following papers can be considered to make the references more comprehensive.

E. Adel, S. El-sappagh, M. Elmogy, S. Barakat and K. Kwak, “A fuzzy ontological infrastructure for semantic interoperability in distributed electronic health record,” Intelligent Automation & Soft Computing, vol. 26, no.2, pp. 237–251, 2020.

Bin Pu, Kenli Li, Shengli Li, Ningbo Zhu: Automatic Fetal Ultrasound Standard Plane Recognition Based on Deep Learning and IIoT. IEEE Trans. Ind. Informatics 17(11): 7771-7780 (2021)

M. A. Khan, S. Abbas and K. M. Khan, “Intelligent forecasting model of Covid-19 novel coronavirus outbreak empowered with deep extreme learning machine,” Computers, Materials & Continua, vol. 64, no. 3, pp. 1329–1342, 2020.

M. Shorfuzzaman and M. Masud, “On the detection of Covid-19 from chest x-ray images using cnn-based transfer learning,” Computers, Materials & Continua, vol. 64, no. 3, pp. 1359–1381, 2020.

J. Wang, Y. Yang, T. Wang, R. Sherratt, J. Zhang. Big Data Service Architecture: A Survey. Journal of Internet Technology, 2020, 21(2): 393-405

M. Shorfuzzaman and M. Masud, “On the detection of Covid-19 from chest x-ray images using cnn-based transfer learning,” Computers, Materials & Continua, vol. 64, no. 3, pp. 1359–1381, 2020.

J. Zhang, S. Zhong, T. Wang, H.-C. Chao, J. Wang. Blockchain-Based Systems and Applications: A Survey. Journal of Internet Technology, 2020, 21(1): 1-14

N. Mualla, E.H. Houssein and M. R. Hassan, “Dental age estimation based on x-ray images,” Computers, Materials & Continua, vol. 62, no. 2, pp. 591–605, 2020.

Author Response

Thank you for the helpful recommendations. Our team sincerely hopes that the attached revised file satisfactorily addresses them to justify publication (with point-by-point response in CAPS below to your specific recommendations and tracked changes included in the attached file).

1. This paper is a case study for Artificial Intelligence-augmented analysis. For the challenges, have authors investigated the solutions, this point can be clearer in the revision. ALL SECTIONS REVISED.

2. The discussions on other areas how people deal with the same issues can be covered. DISCUSSION REVISED.

3. For the evaluation, the dataset/algorithm choosing reasons, the detailed platform configurations and the discussion on other untested datasets should be introduced in the revised paper. METHODS REVISED.

4. Some abbreviations (e.g., AI) are not explained where they first appear.  ALL SECTIONS REVISED AS APPLICABLE.

5. Please add a notation table to explain all used math symbols for easy understanding.  SPECIFICATIONS CONFIRMED. 

6. Please go through the paper carefully and double check whether the right template are used. Correct some errors and formatting issues (e.g., “ and nations)[7,8].” -> “ and nations) [7,8].” ? The necessary space is likely missing). REVISED ALL SECTIONS AS APPLICABLE.

7. Some references lack the necessary information (e.g., [9]), please provide all information according to the right template. REVISIONS REVISED.

8. Resize some oversized figures (e.g., Figure 2). REVISED.

9. The writing of this paper must be well improved, some sentences are too long to understand. ALL SECTIONS REVISED.

10. Make the References more comprehensive, besides Artificial Intelligence-augmented analysis, some other promising scenarios (e.g., Big data, other IoT systems) can be covered in this work. If the above related work can be discussed, it can strongly improve the research significance. For the improvement, the following papers can be considered to make the references more comprehensive. DISCUSSION REVISED.

Round 2

Reviewer 2 Report

Revised paper is much better and can be accepted now.